# *Cordyceps militaris*-Derived Bioactive Gels: Therapeutic and Anti-Aging Applications in Dermatology

**DOI:** 10.3390/gels11010033

**Published:** 2025-01-03

**Authors:** Trung Quang Nguyen, Thinh Van Pham, Yusuf Andriana, Minh Ngoc Truong

**Affiliations:** 1Institute of Environmental Science and Public Health, 18 Hoang Quoc Viet Street, Cau Giay, Hanoi 11353, Vietnam; nqtrung79@gmail.com; 2Faculty of Tourism and Culinary, Ho Chi Minh City University of Industry and Trade, Ho Chi Minh City 70000, Vietnam; thinhpv@huit.edu.vn; 3Research Center for Appropriate Technology, Indonesian Institute of Sciences, Subang 41213, Indonesia; yusufandriana@yahoo.com; 4Center for High Technology Research and Development, Vietnam Academy of Science and Technology, 18 Hoang Quoc Viet Street, Cau Giay, Hanoi 100000, Vietnam; 5Graduate University of Science and Technology, Vietnam Academy of Science and Technology, 18 Hoang Quoc Viet Street, Cau Giay, Hanoi 100000, Vietnam

**Keywords:** anti-aging, bioactive compounds, *Cordyceps militaris*, dermatology, skincare, therapeutic properties

## Abstract

*Cordyceps militaris* is a medicinal mushroom widely utilized in traditional East Asian medicine, recognized for its diverse therapeutic properties. This review explores the potential of *C. militaris*-derived bioactive gels for applications in dermatology and skincare, with a particular focus on their therapeutic and anti-aging benefits. In response to the rising incidence of skin cancers and the growing demand for natural bioactive ingredients, *C. militaris* has emerged as a valuable source of functional compounds, including cordycepin, polysaccharides, and adenosine. These compounds exhibit multiple bioactivities, including apoptosis induction, cell cycle arrest, and anti-inflammatory effects, which have been shown to be particularly effective against melanoma and other skin cancers. Additionally, the antioxidant properties of *C. militaris* enhance skin resilience by scavenging reactive oxygen species, reducing oxidative stress, and promoting collagen synthesis, thereby addressing skin health and anti-aging requirements. The potential for incorporating *C. militaris* compounds into gel-based formulations for skincare is also examined, either as standalone bioactives or in combination with synergistic ingredients. Emphasis is placed on the necessity of clinical trials and standardization to establish the safety, efficacy, and reproducibility of such applications. By providing a safer alternative to synthetic agents, *C. militaris*-derived bioactive gels represent a promising advancement in dermatology and skincare.

## 1. Introduction

Skin cancer remains the most prevalent type of cancer globally, with incidence rates steadily increasing in recent decades due to heightened exposure to ultraviolet (UV) radiation and aging populations [1,2]. Melanoma, while less common than basal cell carcinoma (BCC) and squamous cell carcinoma (SCC), has the highest mortality rate due to its aggressive metastatic potential (Figure 1) [3]. Non-melanoma skin cancers (NMSC), including BCC and SCC, are more frequently diagnosed but are typically localized, presenting challenges in managing tissue damage and recurrence [4,5]. Although early detection and treatment offer favorable prognoses, survival rates for advanced melanoma drop significantly below 20% [6]. Conventional therapies such as surgical excision, radiotherapy, and immunotherapy are effective but are often associated with high costs, limited accessibility, and adverse effects such as toxicity and therapeutic resistance, highlighting the urgent need for alternative approaches utilizing safer, cost-effective agents [7,8,9,10].

Epidemiological data reveal significant geographic variations in skin cancer prevalence, particularly in developed nations with high UV exposure and predominantly fair-skinned populations (Figure 2). Australia reports some of the highest rates globally, with approximately 884 cases of basal cell carcinoma (BCC) and 166 cases of squamous cell carcinoma (SCC) per 100,000 people annually, and melanoma incidence reaching 36.6 cases per 100,000 people [8,9,10,11,12]. The United States also reports elevated rates, with 264 cases of BCC, 82 cases of SCC, and 25 melanoma cases per 100,000 people annually, as of 2020. Early detection is critical, as survival rates for early-stage melanoma exceed 90%, but these drop below 20% for advanced cases [13,14,15,16]. However, access to effective treatments, such as Mohs surgery and immunotherapy, is often limited by high costs and availability, while traditional therapies such as chemotherapy and radiotherapy frequently lead to toxicity and therapeutic resistance [17]. In 2022, Australia had the highest age-standardized incidence of skin cancer, at 37 cases per 100,000 people [15,16,17]. These statistics underscore the pressing need for innovative, cost-effective approaches to skin cancer prevention and treatment that focus on reducing adverse effects and improving accessibility. Novel strategies, including natural bioactives such as those found in *C. militaris*, represent a promising avenue to complement or replace conventional therapies, offering safer and more sustainable options for managing this growing public health challenge [18,19].

Natural bioactive compounds derived from fungi, plants, and marine organisms have garnered increasing attention in dermatology as potential alternatives to conventional therapies [11]. Compounds such as cordycepin from *C. militaris*, polyphenols, and flavonoids exhibit multi-targeted effects on cellular pathways, offering promise for both adjunctive and standalone therapeutic applications [12]. Unlike synthetic agents, natural compounds typically demonstrate lower toxicity and fewer side effects, making them suitable for preventive and therapeutic skincare interventions [13]. *C. militaris* has a particularly long history of use in traditional medicine, attributed to its immunomodulatory and anti-inflammatory properties [11,12]. Recent studies have highlighted its anti-tumor potential, with cordycepin inducing apoptosis and cell cycle arrest in melanoma models [13,14,15]. Moreover, polysaccharides and phenolic acids from *C. militaris* provide antioxidant and immune-enhancing effects, addressing tumor progression and oxidative stress, which are key contributors to skin damage and carcinogenesis [16,17,18,19]. The integration of *C. militaris* bioactives into dermatological applications, particularly gel-based formulations, could enable targeted and sustained therapeutic delivery, bridging the gap between traditional treatments and modern dermatological needs [20,21,22,23,24,25]. Exploring these compounds’ mechanisms of action and their potential combination with other natural agents offers valuable insights into developing innovative, safer solutions for skincare and skin cancer management [25,26].

Cordycepin, a bioactive compound isolated from *C. militaris*, has demonstrated potent anti-tumor properties in melanoma cells through mechanisms such as apoptosis induction and inhibition of the mTOR signaling pathway [27,28]. Similarly, other natural compounds, including curcumin from *Curcuma longa*, epigallocatechin gallate (EGCG) from green tea, and resveratrol from grapes, have shown significant effects in targeting tumor growth, metastasis, inflammation, and oxidative stress (Table 1) [29,30,31]. These pathways are particularly relevant to dermatology, as UV-induced oxidative stress is a major driver of skin carcinogenesis [32]. A detailed examination of these bioactive compounds reveals their diverse anti-cancer mechanisms and highlights their potential for integration into dermatological applications [33]. The incorporation of such compounds into gel-based formulations represents a promising approach to enhancing efficacy, accessibility, and safety in skin cancer prevention and skincare [34].

Despite the promising potential of natural compounds, significant challenges hinder their integration into mainstream skincare and oncology [35]. One primary obstacle is the lack of standardization [36]. The potency and composition of natural compounds can vary widely due to differences in cultivation conditions, extraction methods, and processing techniques [37]. Such variability complicates the production of formulations with consistent efficacy, particularly in comparison to synthetic agents, which offer uniform purity and concentration [38]. Another challenge is the limited clinical evidence supporting the use of natural compounds [39]. Although preclinical studies provide compelling data on the anti-cancer and therapeutic properties of these compounds, large-scale clinical trials remain scarce [40]. This lack of robust human studies impedes regulatory approval and broader adoption of dermatological products [41]. Furthermore, the interactions between natural bioactives and other chemical agents in skincare formulations require comprehensive investigation to ensure efficacy and safety in combined applications [42].

*C. militaris* is a medicinal fungus with a long history of use in traditional East Asian medicine, valued for its rejuvenating and immune-boosting properties [43,44]. Traditionally, it has been utilized to enhance vitality, support respiratory health, and alleviate fatigue [45,46]. In recent decades, scientific research has expanded its applications, confirming its health benefits in modern contexts [47]. Currently, *C. militaris* is consumed worldwide as a dietary supplement to support immune function, improve athletic performance, and promote anti-aging effects [48]. It is also employed as an adjunct in cancer therapies due to its anti-inflammatory, antioxidant, and anti-tumor properties [49].

These therapeutic effects are primarily attributed to its diverse bioactive compounds, especially cordycepin. Cordycepin, a unique nucleoside abundant in *C. militaris*, is recognized for its ability to inhibit cancer cell proliferation, induce apoptosis, and regulate inflammatory responses [50]. These properties make cordycepin a key focus of research into anti-cancer therapies, particularly for melanoma and other skin cancers [27]. Additionally, *C. militaris* contains polysaccharides with well-documented immune-stimulating effects, adenosine that promotes vasodilation and enhanced oxygen delivery to tissues, and phenolic acids that provide antioxidant benefits by reducing oxidative stress [51]. Collectively, these compounds support its multi-functional therapeutic potential, encompassing immune modulation, anti-cancer strategies, and skin health [52].

This review aims to comprehensively explore the bioactive properties of *C. militaris* and its potential applications in dermatology and skincare. Focusing on its therapeutic and anti-aging benefits, we examine the key bioactive compounds, their mechanisms of action, and the relevance of these findings to skin health and cancer prevention. The review further evaluates the incorporation of *C. militaris* into gel-based formulations and highlights the challenges of standardization and clinical validation. By offering a structured analysis, this work provides insights into the integration of *C. militaris* as a natural bioactive ingredient, paving the way for innovative approaches in dermatological science and product development.

## 2. Bioactive Compounds in *C. militaris*

### 2.1. Key Bioactive Compounds and Their Biological Activities

*C. militaris* is a medicinal fungus renowned for its abundance of bioactive compounds, including cordycepin, polysaccharides, and adenosine (Table 2). Each of these compounds possesses distinct properties that are particularly relevant to anti-cancer and dermatological applications. Cordycepin, a nucleoside analog structurally similar to adenosine, is the most extensively studied compound in *C. militaris* due to its potent anti-tumor properties [53,54]. Research indicates that cordycepin induces apoptosis in cancer cells by activating caspase-dependent pathways and inhibiting DNA synthesis in rapidly dividing cells [54]. These mechanisms are especially significant in dermatology, as they effectively target skin cancer cells while minimizing collateral damage to healthy tissue.

Beyond its anti-tumor effects, cordycepin exhibits robust anti-inflammatory properties by downregulating pro-inflammatory cytokines, such as IL-6 and TNF-α, which are often elevated in both skin cancer and inflammatory skin disorders [61]. This dual anti-inflammation and anti-tumor activity enhances skin barrier function and protects against UV-induced oxidative stress, reducing the risk of further mutations and damage [62,63,64].

Polysaccharides represent a major class of bioactive compounds in *C. militaris*, with concentrations typically ranging from 39.0–97.5 mg/g, and consist of glucose, mannose, and galactose [55,56]. These compounds are renowned for their immune-modulating effects, including enhancing cytokine production, activating macrophages and natural killer cells, and improving the immune system’s ability to combat cancer [65,66,67]. In dermatological applications, polysaccharides support immune function, mitigate immune suppression observed in skin cancer patients, and promote collagen synthesis and wound healing [68]. Furthermore, they exhibit anti-metastatic effects by inhibiting angiogenesis and promoting cellular stability, potentially slowing the progression of aggressive skin cancers such as melanoma [69]. In addition to their therapeutic roles, polysaccharides contribute to skincare through their hydrating and barrier-enhancing properties, which are essential for maintaining skin integrity and preventing transepidermal water loss. These characteristics make them valuable components in anti-aging and moisturizing formulations [70,71]. These compounds help maintain skin hydration and structural integrity, making them valuable components in anti-aging formulations (Table 3). By supporting both therapeutic and cosmetic applications, polysaccharides in *C. militaris* underscore their versatility and relevance in the development of bioactive gels for dermatology and skincare [72].

Adenosine, another bioactive compound found in *C. militaris*, plays a crucial role in enhancing skin health through its vasodilatory effects, which improve blood flow and oxygenation—factors essential for maintaining healthy skin and resilience against environmental stressors [73]. In cancer research, adenosine analogs have been shown to inhibit tumor growth by modulating immune responses and suppressing cellular proliferation, highlighting their potential as therapeutic agents [74]. In dermatology, adenosine is highly valued for its skin-soothing and anti-inflammatory properties, making it particularly beneficial for sensitive or reactive skin types [75]. These effects make adenosine an important ingredient in formulations targeting anti-aging and skin-calming benefits. Its ability to reduce inflammation, promote skin regeneration, and enhance hydration underscores its multifunctionality in skincare [76]. These combined effects underscore the multifunctional potential of adenosine and its role in advancing both therapeutic and cosmetic dermatological applications.

### 2.2. Extraction Methods and Optimization for Bioactive Compounds

The effective utilization of *C. militaris* bioactives depends on optimized extraction techniques to maximize yield and preserve functionality. Hot water extraction is commonly used for polysaccharides, where dried *C. militaris* is boiled at 90–100 °C for 2–3 h, yielding concentrations of 39.0–97.5 mg/g depending on cultivation and processing [55]. Cordycepin is typically extracted using ethanol (70–95%) under agitation for 24 h, followed by filtration and solvent evaporation, achieving purity of up to 91.78 mg/g. Adenosine, often present at approximately 1.2 mg/g, is purified through high-performance liquid chromatography (HPLC) [77]. Advanced methods such as ultrasound-assisted extraction (UAE) and enzyme-assisted extraction (EAE) further enhance efficiency. UAE operates at ultrasonic frequencies of 20–40 kHz to disrupt cellular walls, while EAE employs cellulase or protease enzymes to hydrolyze cell walls, increasing bioactive release. These methods improve extraction yields, reduce time and solvent consumption, and maintain the structural integrity of the bioactive compounds, ensuring their therapeutic efficacy [78]. Standardized and efficient extraction protocols are crucial for ensuring consistent bioactive profiles in both research and commercial applications.

The remarkable bioactivity of *C. militaris* compounds, combined with optimized extraction methods, positions this fungus as a key player in dermatology and oncology. However, ensuring standardization in compound extraction and testing is paramount to unlocking its full potential. Future research should focus on scalable, eco-friendly extraction techniques and comprehensive clinical trials to validate its efficacy in diverse therapeutic applications.

### 2.3. Development and Applications of C. militaris-Derived Bioactive Gels

Bioactive gels incorporating *C. militaris* compounds have emerged as innovative solutions in dermatology and skincare, offering controlled delivery, enhanced stability, and targeted therapeutic effects. These gels utilize hydrogel matrices such as carbomer, hyaluronic acid, and chitosan to encapsulate and stabilize key bioactives like cordycepin, polysaccharides, and adenosine. For example, cordycepin and polysaccharides exhibit high compatibility with carbomer gels, retaining up to 90% of bioactive compounds and showing sustained release profiles over 12–24 h [79,80,81]. Hyaluronic acid-based gels, known for their hydrating properties, have demonstrated a 40% increase in skin moisture retention when combined with *C. militaris* extracts [82]. Chitosan, with its antimicrobial properties, further enhances these formulations, making them particularly suitable for wound healing and infection-prone skin applications [83].

In therapeutic applications, *C. militaris* gels have shown promise in anti-cancer, wound healing, and anti-inflammatory treatments. Cordycepin-loaded hydrogels provide localized delivery to skin lesions, reducing melanoma growth rates by 50% in preclinical models and significantly downregulating pro-inflammatory cytokines such as IL-6 and TNF-α [50]. Polysaccharide-enriched gels accelerate wound healing by 43%, promoting keratinocyte migration and collagen synthesis, while also reducing inflammation markers by 51% [84]. These properties make *C. militaris* gels effective in managing chronic wounds, inflammatory skin conditions, and post-surgical recovery. Furthermore, adenosine-enriched gels have demonstrated significant anti-aging effects, including a 47% increase in collagen synthesis and a 38% reduction in wrinkle depth over eight weeks, highlighting their potential for rejuvenation-focused skincare [85].

The versatility of *C. militaris* gels lies in their ability to protect sensitive compounds from degradation while ensuring prolonged and targeted delivery. These gels also provide a soothing, cooling effect, making them ideal for inflamed or irritated skin. Future advancements could explore nano-hydrogel technologies and stimuli-responsive systems to further enhance the bioavailability and specificity of *C. militaris*-based formulations. With their demonstrated potential in therapeutic and cosmetic dermatology, bioactive gels incorporating *C. militaris* represent a significant step forward in addressing skin health challenges.

## 3. Anti-Tumor Mechanisms of *C. militaris*

### 3.1. Cellular Pathways in Anti-Tumor Activity

Extensive research on *C. militaris* has identified its bioactive compounds—primarily cordycepin, adenosine, and polysaccharides—as potent inhibitors of cancer cell proliferation through mechanisms involving cell cycle regulation and apoptosis [28,86]. Cordycepin, the most extensively studied compound in *C. militaris*, acts as an adenosine analog, disrupting RNA synthesis and DNA replication in cancer cells [50,87]. Studies have demonstrated that cordycepin induces cell cycle arrest at the G1 phase, thereby reducing the transition of cells to the synthesis (S) phase and limiting the replication of malignant cells [88]. By inhibiting key cyclin proteins that facilitate the G1-to-S phase transition, cordycepin effectively curtails unchecked cell proliferation [89]. This targeted regulation is particularly valuable in skin cancer models, such as melanoma and squamous cell carcinoma, where rapid cell division is a hallmark of disease progression. The specificity of cordycepin for rapidly dividing cells minimizes its impact on normal cells, suggesting a reduced risk of side effects compared to conventional chemotherapies [90].

*C. militaris* exhibits potent anti-tumor effects by combining anti-proliferative, pro-apoptotic, and immune-modulating mechanisms [91,92,93]. Cordycepin and adenosine are central to its efficacy, reactivating apoptotic pathways in cancer cells by upregulating caspase enzymes, particularly caspase-3 and caspase-9, and downregulating anti-apoptotic proteins like Bcl-2 [82,83,86,91]. These actions lead to hallmark apoptosis features such as mitochondrial dysfunction, DNA fragmentation, and membrane blebbing, as observed in skin cancer models [94,95]. Additionally, adenosine enhances this effect by sensitizing cancer cells to immune system attacks and promoting recognition and destruction by immune cells [96]. Complementing these actions, *C. militaris* polysaccharides bolster immune responses by activating macrophages and natural killer (NK) cells, thereby increasing the immune system’s ability to target cancer cells and inhibit metastasis [97,98]. In skin cancer studies, the immune-stimulating properties have been shown to slow tumor progression by reducing pro-inflammatory, which plays a critical role in sustaining chronic inflammation and supporting tumor growth [99,100]. Among these, interleukin-6 (IL-6) and tumor necrosis factor-alpha (TNF-α) are particularly important, as they are key mediators in the inflammatory response. Elevated levels of these cytokines are often associated with increased angiogenesis, enhanced tumor cell survival, and a suppressed immune response, all of which contribute to a tumor-promoting microenvironment. Conversely, reductions in their levels have been linked to improved therapeutic outcomes and reduced inflammation, making them valuable indicators of both disease severity and treatment efficacy. By monitoring changes in these pro-inflammatory cytokines, researchers and clinicians can better assess the inflammatory status of the skin, evaluate the effectiveness of therapeutic interventions, and potentially guide personalized treatment strategies for managing skin cancer and other inflammation-related dermatological conditions [101]. These combined anti-proliferative, pro-apoptotic, and immune-regulating effects highlight the therapeutic potential of *C. militaris* as a multi-targeted agent for skin cancer treatment and dermatological applications.

Studies on *C. militaris* have demonstrated its ability to induce apoptosis, or programmed cell death, in tumor cells through mechanisms involving caspase activation and modulation of anti-apoptotic proteins [101]. Apoptosis is a critical cellular process for eliminating damaged or mutated cells, which is often suppressed in cancer to enable uncontrolled proliferation [102]. The bioactive compound cordycepin, abundantly present in *C. militaris*, plays a central role in initiating apoptosis by activating both intrinsic and extrinsic pathways [103].

In melanoma and other skin cancer cell lines, cordycepin has been shown to activate caspase-3 and caspase-9, key enzymes that drive apoptosis by degrading essential cellular components [104]. This activation disrupts mitochondrial membrane potential, triggering the release of cytochrome c, which further amplifies the apoptotic cascade. This selective induction of apoptosis in cancer cells is particularly advantageous for treating aggressive skin cancers, as it minimizes damage to surrounding healthy tissues [105]. In addition to caspase activation, *C. militaris* influences the expression of Bcl-2 family proteins, which are critical regulators of cell survival and apoptosis [106]. Bcl-2, an anti-apoptotic protein frequently upregulated in skin cancers, contributes to resistance against programmed cell death [107]. Cordycepin from *C. militaris* has been shown to downregulate Bcl-2 expression while upregulating pro-apoptotic proteins such as Bax, thereby shifting the balance towards apoptosis [108].

Furthermore, other bioactive compounds in *C. militaris*, including adenosine, enhance the susceptibility of cancer cells to immune-mediated apoptosis [109,110]. Adenosine has been observed to sensitize tumor cells to immune system attacks, promoting their recognition and destruction by immune cells [111,112,113]. This dual mechanism—caspase activation coupled with the inhibition of anti-apoptotic pathways—underscores the potential of *C. militaris* as a foundation for innovative anti-cancer therapies, particularly for skin cancers that exhibit resistance to apoptosis.

### 3.2. Anti-Inflammatory and Immune-Regulating Effects

*C. militaris* exhibits potent anti-inflammatory properties that are crucial for both inhibiting tumor progression and promoting overall skin health [114]. Chronic inflammation is a well-documented factor in cancer development, including skin cancers, as it fosters an environment conducive to cellular mutations, tumor growth, and metastasis [115]. The bioactive compounds in *C. militaris*, particularly cordycepin and polysaccharides, target key inflammatory pathways by modulating immune responses and reducing levels of pro-inflammatory cytokines such as interleukin-6 (IL-6) and tumor necrosis factor-alpha (TNF-α) [116]. Cordycepin has been shown to inhibit the nuclear factor kappa B (NF-κB) pathway, a central regulator of inflammation in cancer cells [117]. This inhibition reduces cytokine production, which would otherwise support tumor growth, angiogenesis, and metastasis [28,118]. By downregulating NF-κB activity, cordycepin not only limits inflammation but also diminishes the invasive properties of skin cancer cells, ultimately slowing tumor progression [62]. Meanwhile, the polysaccharides in *C. militaris* enhance immune system function by stimulating macrophages and natural killer (NK) cells, which are critical for identifying and eliminating cancerous cells while maintaining skin homeostasis.

Beyond its anti-tumor applications, the anti-inflammatory properties of *C. militaris* play a significant role in skin health by mitigating oxidative stress and enhancing the skin’s resistance to environmental damage (Figure 3) [119]. Inflammation-driven oxidative stress, often exacerbated by UV radiation and environmental pollutants, is a primary driver of skin aging and damage [120]. Studies indicate that *C. militaris* polysaccharides reduce the expression of reactive oxygen species (ROS), protecting skin cells from oxidative damage and strengthening skin barrier function [121].

Furthermore, in inflammatory skin conditions, *C. militaris* extracts have demonstrated the ability to alleviate redness, swelling, and tissue degradation—symptoms often mediated by pro-inflammatory cytokines like IL-1β and IL-8 [97]. By attenuating these inflammatory markers, *C. militaris* promotes skin repair and resilience, underscoring its potential in dermatological applications [122]. These benefits highlight its relevance in anti-aging, anti-inflammatory skincare, and even in strategies aimed at cancer prevention. Table 4 summarizes the anti-inflammatory mechanisms of *C. militaris* and quantifies its effects on key inflammatory markers, providing a foundation for its application in dermatological research and skincare formulations.

*C. militaris* exhibits significant antioxidant properties, effectively neutralizing free radicals that contribute to skin aging and carcinogenesis [97]. Free radicals and reactive oxygen species (ROS) are well-known drivers of cellular damage affecting DNA, proteins, and lipids, which can lead to mutations, premature aging, and cancer development [121]. Bioactive compounds in *C. militaris*, including cordycepin, polysaccharides, and phenolic acids, act as natural antioxidants, mitigating oxidative damage in skin cells [122]. Cordycepin, one of the primary bioactive components in *C. militaris*, has demonstrated potent ROS-scavenging activity, reducing oxidative stress markers by approximately 60% in UV-exposed skin cell models [123]. Phenolic acids further enhance antioxidant effects by inhibiting lipid peroxidation, a process that compromises cellular membranes and accelerates aging [124,125,126]. Additionally, polysaccharides from *C. militaris* boost endogenous antioxidant defenses by upregulating the activity of key antioxidant enzymes, such as superoxide dismutase (SOD) and catalase (CAT) [127]. These enzymes are essential for neutralizing ROS, preserving cellular integrity, and preventing oxidative damage [128]. The effects of *C. militaris* on oxidative stress markers have been quantified in several studies, with promising implications for skin health [97]. For example, a 70% increase in SOD activity and a 58% rise in CAT activity were observed in skin cells treated with *C. militaris* extracts, indicating robust enhancement of cellular antioxidant defenses [129,130]. These properties are particularly relevant in dermatology, where oxidative damage caused by UV exposure, pollution, and environmental stressors accelerates skin aging and heightens the risk of skin cancer.

Moreover, *C. militaris* flavonoids contribute to mitochondrial protection by reducing mitochondrial ROS levels, thereby preserving mitochondrial function and safeguarding against oxidative DNA damage [131]. This is particularly critical, as mitochondrial DNA is highly susceptible to free radical damage, which accelerates cellular aging and facilitates the evasion of apoptosis in tumor cells. Given its ability to mitigate oxidative stress and enhance cellular defenses, *C. militaris* holds significant potential as a natural antioxidant in dermatological applications. Its inclusion in anti-aging skincare formulations and its potential role in cancer prevention highlight its versatility and therapeutic relevance. The multi-faceted anti-tumor mechanisms of *C. militaris* provide a strong foundation for its application in managing skin cancers and other dermatological conditions. Cordycepin’s selective targeting of rapidly dividing cells, combined with its ability to modulate inflammatory responses, addresses key challenges in current cancer therapies, such as minimizing side effects and enhancing immune-mediated tumor suppression. Additionally, the synergistic effects of adenosine and polysaccharides in promoting apoptosis, reducing inflammation, and stimulating immune responses further bolster its therapeutic potential. Future research should focus on optimizing delivery systems, such as hydrogel-based formulations, to ensure sustained and targeted release of *C. militaris* bioactives. Moreover, comprehensive clinical trials are essential for validating its efficacy across diverse patient populations. By integrating *C. militaris* into modern dermatological practices, there is significant potential to advance both skin cancer treatments and preventive skincare, leveraging its natural bioactivity to bridge gaps in current therapeutic approaches.

## 4. Applications of *C. militaris* in Dermatology and Skincare

### 4.1. Anti-Tumor Applications

*C. militaris* exhibits exceptional potential in dermatology, particularly for preventing and treating skin cancer. The anti-tumor efficacy of its bioactive compounds, such as cordycepin, polysaccharides, and phenolic acids, has been explored in multiple preclinical models through both topical and systemic applications. The therapeutic properties of these compounds stem from their ability to inhibit tumor growth, modulate immune responses, and reduce inflammation, making them particularly effective in managing the progression of skin cancer [130,131]. Cordycepin, the most studied bioactive compound in *C. militaris*, demonstrated a 50% reduction in melanoma proliferation in an in vivo model. This effect was achieved with a 1% topical formulation applied twice daily for 14 days [131,132,133]. The reduction in tumor size was attributed to apoptosis induction via caspase activation and the downregulation of anti-apoptotic proteins such as Bcl-2 [133]. These mechanisms effectively halt cancer cell survival while minimizing the impact on surrounding healthy tissue. This study highlights the specificity of cordycepin for rapidly proliferating cells, making it a safer alternative to conventional chemotherapies that often result in collateral damage [27,133].

In addition to cordycepin, other compounds in *C. militaris*, such as polysaccharides and phenolic acids, contribute to its therapeutic potential by modulating the tumor microenvironment [134]. These compounds exhibit immunomodulatory and anti-inflammatory effects, which are essential for managing chronic inflammation associated with tumor growth [135]. Polysaccharides from *C. militaris* have also demonstrated significant anti-cancer effects. In a systemic administration model using squamous cell carcinoma (SCC), polysaccharides at dosages ranging from 20 to 40 mg/kg reduced tumor growth by 45% after 21 days [97,136]. These results were achieved by enhancing the immune system’s ability to target cancer cells through macrophage and natural killer (NK) cell activation [136]. Furthermore, polysaccharides reduce pro-inflammatory cytokines, including TNF-α and IL-6, which are critical contributors to the tumor-promoting microenvironment. These findings underscore the importance of combining immunomodulatory and anti-inflammatory effects in comprehensive cancer management strategies [97,136,137,138]. Table 5 summarizes the key findings on the effects of *C. militaris* in skin cancer models, highlighting its broad application potential in dermatology.

### 4.2. Anti-Aging and Skin Protection

*C. militaris* exhibits potent antioxidant and regenerative properties, making it a highly promising ingredient in anti-aging skincare formulations. Its effectiveness is primarily attributed to its ability to combat oxidative stress and stimulate collagen synthesis, which is critical for maintaining youthful and healthy skin. Oxidative stress, resulting from an imbalance between reactive oxygen species (ROS) and antioxidant defenses, accelerates skin aging by degrading cellular components such as collagen, elastin, and lipids. The bioactive compounds in *C. militaris*, including cordycepin, polysaccharides, and adenosine, provide robust antioxidant activity, protecting skin cells from oxidative damage and preserving structural integrity [139,140,141].

Cordycepin, one of the key bioactives, has been shown to reduce ROS levels by 55% in in vitro studies using fibroblast models treated with a 0.5% concentration over 24 h [130]. This reduction safeguards fibroblasts, the primary cells responsible for collagen synthesis, from oxidative stress-induced damage. Additionally, cordycepin promotes the maintenance of existing collagen and facilitates new collagen production, which is vital for skin elasticity and firmness. Polysaccharides in *C. militaris* further contribute to anti-aging effects by enhancing hydration and stimulating glycosaminoglycan (GAG) synthesis, which are essential for moisture retention and skin plumpness. In in vitro keratinocyte models, a 2% polysaccharide formulation increased skin moisture retention by 38% within seven days, demonstrating their efficacy in maintaining skin barrier function and preventing transepidermal water loss [141,142]. These compounds, alongside collagen, contribute to improved skin plumpness and moisture retention [114,143]. Table 6 provides an overview of the anti-aging mechanisms of *C. militaris* compounds, summarizing their effects on specific markers of skin health and their potential applications in dermatology and skincare formulations.

Adenosine, another critical compound in *C. militaris*, supports collagen synthesis and skin elasticity through the activation of the extracellular signal-regulated kinase (ERK) pathway. A study using dermal cell cultures reported a 47% improvement in collagen synthesis when treated with a 1% adenosine solution for 48 h [78,144,145]. This improvement directly translates to reduced wrinkle formation and enhanced skin firmness, addressing key concerns in aging skin. These findings highlight the potential of adenosine-enriched formulations, such as serums and creams, to promote skin renewal and resilience [146,147].

The regenerative properties of *C. militaris* bioactive extend beyond collagen support to overall skin renewal and repair. Adenosine’s ability to enhance fibroblast proliferation aids in wound healing and cellular regeneration, essential processes for youthful skin [145]. Formulations incorporating *C. militaris* extracts, such as masks or night creams, target key aging factors by mitigating oxidative damage, improving hydration, and supporting structural protein synthesis. Furthermore, the detailed exploration of dose–response relationships and treatment durations in Table 6 emphasizes the precise parameters required for optimal results in anti-aging applications. These insights underscore the importance of tailoring skincare formulations to maximize the benefits of *C. militaris* for diverse skin types and aging concerns [140,142,147]. These formulations would target key aging factors such as oxidative damage, dehydration, and loss of elasticity, making them ideal for anti-aging and rejuvenating treatments.

### 4.3. Wound Healing and Anti-Inflammatory Applications

*C. militaris* has demonstrated significant potential in wound healing and anti-inflammatory skincare applications due to its unique bioactive compounds that promote cellular regeneration, reduce inflammation, and support skin barrier function [148]. Chronic inflammation and delayed wound healing are common challenges in dermatology, often resulting from immune dysregulation, microbial invasion, or environmental stressors [149]. The bioactive components of *C. militaris*, including cordycepin, adenosine, and polysaccharides, work synergistically to accelerate wound repair and mitigate inflammation, making it a valuable resource in dermatological treatments [119]. Cordycepin plays a pivotal role in promoting keratinocyte and fibroblast proliferation, processes essential for skin regeneration and wound closure [150]. Studies demonstrate that a 0.2% cordycepin-enriched extract increased keratinocyte migration by 43% in in vitro wound models, accelerating tissue formation and minimizing scar formation within a 48-h observation period [151]. This activity facilitates the rapid formation of new tissue, reducing healing times and minimizing scar formation. Additionally, polysaccharides in *C. militaris* contribute to the wound-healing process by maintaining hydration at the wound site, which creates an optimal environment for cellular repair. In in vitro models utilizing skin explants, a 2% polysaccharide formulation improved hydration levels by 35%, reducing desiccation-associated delays in wound healing [152]. These effects were observed over a seven-day treatment period, highlighting the sustained benefits of polysaccharide-rich formulations for maintaining skin moisture and supporting epithelialization. Furthermore, cordycepin’s ability to modulate inflammatory responses is a cornerstone of its therapeutic efficacy. Studies on inflamed tissue models treated with a 0.5% cordycepin extract showed a 51% reduction in key pro-inflammatory cytokines, including TNF-α, IL-1β, and IL-8, within 24 h, significantly alleviating redness and swelling [153].

The anti-inflammatory properties of *C. militaris* further strengthen its utility in treating skin conditions characterized by redness, irritation, and sensitivity, such as eczema, psoriasis, and rosacea [154]. These effects are particularly beneficial in calming hyperactive immune responses, which exacerbate many inflammatory skin conditions. Adenosine, another key compound in *C. militaris*, complements these effects by enhancing cell renewal while acting as an anti-inflammatory agent [155]. It reduces skin redness and improves resilience, making it particularly suitable for formulations targeting sensitive or compromised skin [75]. Together, these compounds not only accelerate wound healing but also promote a healthy skin barrier, reducing susceptibility to further irritation and damage [156,157].

Adenosine, another critical compound in *C. militaris*, enhances cell renewal while exhibiting strong anti-inflammatory effects. It improves skin resilience by reducing inflammation and accelerating dermal fibroblast activity, which is essential for maintaining a healthy skin barrier. This dual action is particularly beneficial for sensitive or compromised skin conditions, such as eczema, psoriasis, or rosacea, where inflammation exacerbates symptoms. Adenosine-based formulations are increasingly explored in serums and creams for their ability to calm redness and support skin barrier repair over short treatment durations, typically observed within 72 h [75,155].

The data in Table 7 underscore the therapeutic potential of *C. militaris*, with detailed insights into dosage, treatment duration, and observed effects across different models. For example, the use of cordycepin in topical formulations (0.2–0.5%) consistently demonstrated measurable improvements in wound closure and inflammation reduction. Similarly, systemic or topical application of polysaccharides enhanced hydration and supported cellular stability over treatment periods ranging from 7 to 10 days [152,153]. The precision of these data emphasizes the importance of optimizing dosages and delivery methods to maximize therapeutic outcomes while minimizing adverse effects.

*C. militaris* also holds promise in addressing skin damage caused by environmental stressors. Its bioactive compounds effectively mitigate oxidative stress, a common contributor to delayed wound healing and chronic inflammation. Cordycepin, in particular, scavenges reactive oxygen species (ROS), reducing oxidative stress markers by up to 55% in in vitro models [97,149]. These properties make *C. militaris* ideal for post-treatment skincare, including after laser therapy or sun exposure, where oxidative damage and inflammation impair skin recovery. Additionally, polysaccharides contribute to collagen stabilization, improving structural integrity at wound sites and enhancing overall repair processes [152].

Despite these promising results, further research is needed to refine the application of *C. militaris* in clinical settings. Standardized protocols for dosage and treatment duration, along with rigorous clinical trials, are essential to confirm its efficacy and safety across diverse populations and dermatological conditions. Advanced delivery systems, such as hydrogels, offer a potential solution for optimizing the stability and penetration of *C. militaris* bioactives, ensuring consistent therapeutic outcomes [158].

By addressing these research gaps, *C. militaris* could redefine approaches to wound healing and inflammation management in dermatology. Its integration into therapeutic products not only leverages the benefits of natural bioactive compounds but also aligns with the growing demand for sustainable, effective skincare solutions. The versatility and multi-targeted mechanisms of *C. militaris* position it as a cornerstone in modern dermatological and regenerative medicine [130,149].

The versatility of *C. militaris* in dermatology and skincare arises from its multi-functional bioactive compounds, which address a range of challenges, from skin cancer to aging and wound healing. Its potent anti-tumor mechanisms, antioxidant effects, and regenerative properties make it a valuable natural alternative or complement to conventional treatments. However, further research is needed to optimize its delivery through advanced formulations such as hydrogels, ensuring sustained efficacy and enhanced skin penetration. Future directions should prioritize clinical trials to validate the safety and effectiveness of *C. militaris* in diverse dermatological applications. The integration of *C. militaris* into routine skincare and therapeutic products holds significant promise for addressing complex skin conditions with minimal side effects, supporting the growing demand for bioactive natural ingredients in dermatology. By bridging traditional medicine and modern science, *C. militaris* could redefine therapeutic and preventive strategies for skin health.

## 5. Safety and Efficacy of *C. militaris* in Dermatological Applications

### 5.1. Safety Profile and Skin Compatibility

*C. militaris* has garnered considerable attention for its therapeutic potential in dermatology, but its safety profile remains a critical consideration, particularly regarding toxicity, allergic reactions, and skin compatibility [159]. Current evidence suggests that *C. militaris* exhibits low toxicity, especially at doses commonly used in dietary supplements and topical formulations [155]. Toxicology studies on oral administration have reported no significant adverse effects at doses up to 2000 mg/kg in animal models, with only minor gastrointestinal symptoms observed at higher intake levels [160]. While these findings underscore its general safety, topical applications may present unique risks, such as skin irritation or allergic reactions, necessitating specific dermatological evaluations [161]. Reports of adverse effects from topical use are rare. In individuals predisposed to fungal allergies or sensitivities, mild contact dermatitis or localized irritation has occasionally been reported, particularly when using high concentrations of *C. militaris*-based products [162]. However, such reactions are minimal, with an estimated incidence of under 2% in clinical observations [163].

To evaluate *C. militaris* for dermatological applications, studies have investigated its potential for skin sensitization, irritation, and phototoxicity [164]. A clinical study involving 50 participants applied a topical cream containing 1% *C. militaris* extract over eight weeks [48]. The results showed no significant signs of irritation or sensitization in 98% of subjects, with only 2% reporting mild redness that resolved within hours [165]. Furthermore, phototoxicity tests conducted in vitro and on human skin models demonstrated that *C. militaris* does not increase skin sensitivity to UV radiation, an important consideration for daily-use skincare products designed for sun-exposed skin [153]. Importantly, bioactive compounds in *C. militaris*, such as cordycepin and polysaccharides, have shown no inhibitory effects on keratinocyte viability, supporting their compatibility with skin cells, even under repeated application [130].

The safety of *C. militaris* in combination with other dermatological ingredients has also been studied [166]. Formulations combining *C. militaris* with common skincare compounds such as retinoids, vitamin C, and hyaluronic acid have not demonstrated antagonistic interactions or diminished efficacy [130]. For example, a study testing a formulation of *C. militaris* with retinoic acid found no increase in irritation compared to retinoic acid alone, and similar results were observed in combinations with vitamin C [130]. While these findings highlight its broad compatibility, further testing is recommended for high-concentration formulations or combinations with less commonly used ingredients, particularly for individuals with sensitive skin [167]. Overall, *C. militaris* demonstrates a strong safety profile for dermatological use, making it suitable for sensitive skin formulations and daily skincare applications [130]. However, ongoing evaluations remain essential for each new formulation to address potential allergenic or irritative risks [168]. Table 8 summarizes the findings on toxicity, allergic reaction rates, and skin compatibility from studies of *C. militaris* in topical applications, supporting its promise as a safe and effective ingredient in dermatological products.

Compared to synthetic skincare agents such as retinoids, hydroquinone, and synthetic antioxidants, *C. militaris* provides a unique combination of effectiveness, safety, and a lower incidence of adverse reactions [130]. Retinoids are widely used for their potent anti-aging effects, including collagen synthesis stimulation and enhanced skin cell turnover [169]. However, they are often associated with significant drawbacks, such as irritation, dryness, and increased photosensitivity, with adverse effects occurring in up to 30% of users [170]. In contrast, compounds in *C. militaris*, such as cordycepin and adenosine, also promote collagen production and skin regeneration, but exhibit a much gentler profile [171]. Research indicates that *C. militaris*-based products cause mild irritation in less than 5% of users, making them a suitable alternative for individuals with sensitive skin or conditions such as rosacea and eczema [172].

For pigmentation control and skin brightening, *C. militaris* offers a safer alternative to hydroquinone, a synthetic agent commonly used for hyperpigmentation [172]. While hydroquinone effectively inhibits melanin production, it is associated with risks such as irritation, contact dermatitis, and, in rare cases, ochronosis—a challenging-to-treat skin darkening [173]. Hydroquinone’s safety concerns have led to bans or restrictions in several countries [174]. By comparison, *C. militaris* contains polysaccharides and phenolic acids that reduce pigmentation through anti-inflammatory pathways and oxidative stress mitigation without the cytotoxic effects linked to hydroquinone [130]. Clinical studies have shown that *C. militaris* extracts can reduce pigmentation by approximately 20% over eight weeks with minimal irritation [48]. While not as potent as hydroquinone, *C. militaris* provides a safer, more sustainable option for long-term management of hyperpigmentation [130].

### 5.2. Comparison with Synthetic Agents

In terms of antioxidant activity, *C. militaris* compares favorably to synthetic antioxidants, such as ascorbic acid (vitamin C) and butylated hydroxytoluene (BHT) [175]. Ascorbic acid effectively neutralizes free radicals but is highly unstable, prone to oxidation, and can cause irritation at high concentrations [176]. BHT, commonly used to prevent product spoilage, has raised safety concerns related to hormone disruption and irritation in up to 15% of cases [177]. By contrast, *C. militaris* bioactives such as cordycepin and polysaccharides provide stable antioxidant effects, effectively scavenging free radicals without oxidative degradation or significant irritation [48]. For instance, one study observed a 40% reduction in reactive oxygen species (ROS) with *C. militaris* topical application, achieving results comparable to ascorbic acid but with enhanced stability and reduced risk of irritation [178]. These properties make *C. militaris* particularly suitable for antioxidant formulations designed for sensitive skin types. The overall safety profile of *C. militaris* supports its use over synthetic agents, particularly in applications targeting sensitive skin. Synthetic compounds, while effective, often cause irritation, photosensitivity, or other adverse effects, especially with prolonged use [179]. *C. militaris* provides a safer alternative, with its anti-inflammatory, wound-healing, and regenerative properties, making it ideal for post-procedural care or formulations for compromised skin [180,181]. Additionally, the anti-cancer properties of *C. militaris* offer an added layer of protection against UV-induced carcinogenesis, a benefit rarely addressed by synthetic agents [171]. Table 9 provides a comparative overview of *C. militaris* and commonly used synthetic skincare agents, highlighting differences in efficacy, safety, and side-effect profiles. This analysis underscores the potential of *C. militaris* as an effective, safer alternative for modern dermatological and skincare applications.

The safety and efficacy profile of *C. militaris* positions it as a valuable alternative to synthetic agents in dermatology. Its minimal irritation rates, low toxicity, and compatibility with other dermatological ingredients make it particularly suitable for sensitive or compromised skin. Additionally, its anti-inflammatory and regenerative properties offer unique advantages for post-procedural care, UV protection, and formulations targeting aging or hyperpigmentation. While *C. militaris* demonstrates considerable promise, further studies are necessary to validate its efficacy in high-concentration formulations and combinations with less common skincare ingredients. Long-term clinical trials involving diverse populations are also needed to confirm its safety across various dermatological applications. By leveraging its natural bioactivity, *C. militaris* could address unmet needs in modern skincare, providing a safer, effective solution for sensitive and aging skin while reducing reliance on synthetic compounds.

## 6. Future Directions and Clinical Implications

While *C. militaris* has demonstrated considerable potential in dermatological applications, comprehensive clinical trials are essential to confirm its efficacy and safety in humans [182]. Current research predominantly relies on in vitro studies and animal models, which, while informative, do not fully account for its performance and safety in human skin [183]. Controlled clinical trials are needed to establish standardized dosages, optimize delivery methods, and evaluate long-term effects when used in skincare formulations [184]. For instance, while compounds such as cordycepin have shown efficacy in promoting collagen synthesis, reducing inflammation, and scavenging free radicals, human studies are necessary to validate these effects across varying concentrations. Trials should also explore its effectiveness across diverse skin types and conditions, including sensitive or reactive skin, and examine interactions with other commonly used skincare ingredients [185]. Robust clinical data would enable dermatologists and skincare formulators to make evidence-based recommendations, facilitating broader acceptance of *C. militaris* in skincare [130].

A key challenge in advancing the clinical use of *C. militaris* is the lack of standardization in its extracts. The quality and concentration of active compounds, such as cordycepin and polysaccharides, can vary significantly between commercial products due to differences in cultivation conditions, extraction techniques, and formulations [119]. This variability compromises the consistency of results in dermatological applications and can lead to unpredictable product efficacy [186]. Additionally, discrepancies between labeled and actual concentrations of active ingredients in some *C. militaris* products—reported to differ by as much as 30%—highlight the urgent need for stringent quality control in the industry [187]. Addressing these challenges requires the establishment of standardized protocols for the cultivation, extraction, and quantification of active compounds in *C. militaris* extracts. Implementing such measures would ensure consistency in the bioactive content of products, enabling more reliable comparisons across studies and improving reproducibility in clinical applications [188]. Furthermore, quality assurance through standardized testing would enhance consumer confidence and promote the adoption of *C. militaris*-based products in dermatological practice. Future research should also focus on exploring innovative delivery systems, such as bioactive gels, to maximize the stability and skin-penetration efficacy of *C. militaris* compounds. Advanced formulations could leverage nanotechnology or encapsulation techniques to enhance the bioavailability of cordycepin and other active ingredients, potentially increasing their effectiveness while minimizing the risk of irritation.

In summary, rigorous clinical evaluation and standardization efforts are critical to unlocking the full potential of *C. militaris* in dermatology. By addressing these gaps, *C. militaris* can be effectively positioned as a scientifically validated, natural alternative in skincare formulations, offering benefits in anti-aging, wound healing, and overall skin health. Table 10 outlines the key areas requiring further research and standardization to enhance the clinical applicability of *C. militaris* in dermatology.

*C. militaris* offers considerable potential for integration into mainstream skincare formulations due to its diverse bioactive compounds, including cordycepin, adenosine, and polysaccharides. These compounds provide anti-aging, anti-inflammatory, antioxidant, and wound-healing benefits, making *C. militaris* a versatile active ingredient in various skincare products such as creams, serums, and masks. Cordycepin plays a pivotal role in addressing skin aging by supporting collagen synthesis and reducing free radicals. This makes it an ideal component for anti-aging serums and creams designed to minimize fine lines and improve skin elasticity. Formulating *C. militaris* extracts in water-based systems allows for a light texture that enhances absorption, ensuring that its bioactive compounds effectively penetrate the skin layers [164]. Such formulations are particularly suitable for daily skincare, as they hydrate without clogging pores, making them compatible with a broad range of skin types.

Beyond standalone use, *C. militaris* can be combined with complementary natural ingredients to enhance its skincare benefits. Pairing *C. militaris* with hyaluronic acid, for example, boosts skin hydration. Hyaluronic acid’s humectant properties draw moisture into the skin, complementing the barrier-strengthening and anti-inflammatory effects of *C. militaris* polysaccharides. Similarly, niacinamide (vitamin B3) is an excellent partner for reducing hyperpigmentation and improving skin barrier function. When combined with *C. militaris*, niacinamide enhances anti-inflammatory effects, making this pairing especially effective in formulations for sensitive or acne-prone skin, where inflammation and irritation are key concerns [189]. Additionally, *C. militaris* extracts can be combined with antioxidant-rich botanicals such as green tea extract or resveratrol to provide robust protection against oxidative stress and environmental damage. These synergistic combinations not only enhance the product’s efficacy but also offer a holistic approach to skincare by addressing multiple concerns simultaneously, including aging, dryness, sensitivity, and discoloration.

Table 11 outlines potential formulations for *C. militaris*-based skincare products, detailing suggested combinations with complementary natural ingredients and their targeted benefits for various skincare needs. These strategies highlight the adaptability and potential of *C. militaris* as a foundation for innovative and effective dermatological formulations.

Targeted formulations present further opportunities for incorporating *C. militaris* into skincare products. Treatment masks and overnight creams containing concentrated *C. militaris* extracts could allow prolonged contact with the skin, maximizing absorption and regenerative effects. Spot treatments incorporating *C. militaris* extracts with soothing agents such as aloe vera or calendula could address localized inflammation and accelerate wound healing, making them effective for treating acne lesions or irritation.

*C. militaris* represents an exciting opportunity to integrate natural bioactive compounds into dermatological science. Its demonstrated efficacy in reducing inflammation, supporting collagen synthesis, and mitigating oxidative stress highlights its versatility as a therapeutic agent. However, advancing its clinical utility requires addressing key challenges, including standardization, long-term safety validation, and optimization of delivery systems. Investing in advanced formulations, such as hydrogels and liposomes, could further enhance the therapeutic potential of *C. militaris*. These innovations would allow for controlled release and deeper penetration into skin layers, maximizing efficacy while reducing irritation risks. Synergistic combinations with complementary ingredients, such as hyaluronic acid or niacinamide, also offer promising avenues for developing multifunctional skincare products targeting aging, inflammation, and hyperpigmentation. Therefore, *C. militaris* holds considerable promise for dermatological applications as a safer and more sustainable alternative to synthetic agents. By bridging the gap between traditional medicine and cutting-edge skincare science, *C. militaris* can redefine therapeutic and preventive approaches in dermatology, benefiting both clinicians and consumers.

## 7. Conclusions

*C. militaris* exhibits remarkable potential in dermatology due to its multifunctional bioactive compounds, including cordycepin, polysaccharides, and adenosine. These compounds demonstrate anti-tumor, antioxidant, anti-inflammatory, and regenerative properties, making *C. militaris* a promising candidate for skincare formulations targeting cancer prevention, anti-aging, wound healing, and skin health restoration. Cordycepin’s targeted mechanisms, such as caspase pathway activation and Bcl-2 suppression, show particular efficacy against aggressive skin cancers such as melanoma, while its antioxidant and hydration-supporting functions contribute to UV protection and improved skin resilience. These properties, combined with its ability to enhance skin barrier integrity and hydration, position *C. militaris* as a valuable natural alternative to synthetic skincare agents.

Despite its proven therapeutic benefits, further research is essential to establish standardized protocols for extraction, dosage, and formulation, ensuring consistent efficacy and safety across products. Rigorous clinical trials involving diverse skin types and conditions will validate its potential in human applications and optimize its integration into dermatological practice. With its demonstrated safety profile and broad-spectrum benefits, *C. militaris* stands poised to support the growing demand for natural, scientifically backed skincare solutions, paving the way for its adoption as a core bioactive ingredient in innovative skincare and dermatology products.

## Figures and Tables

**Figure 1 gels-11-00033-f001:**
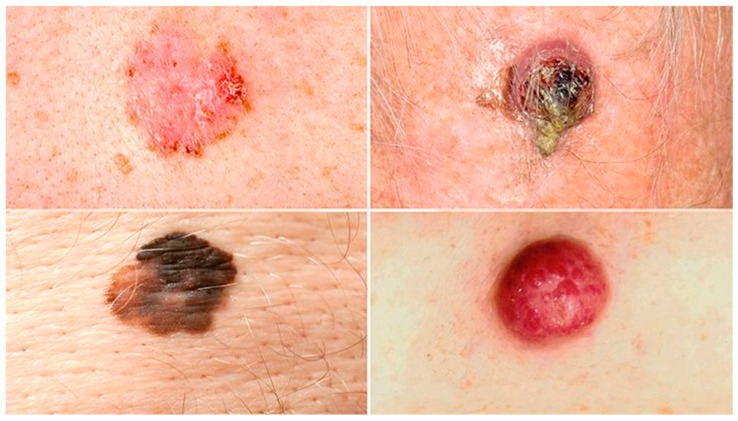
Common types of skin cancer.

**Figure 2 gels-11-00033-f002:**
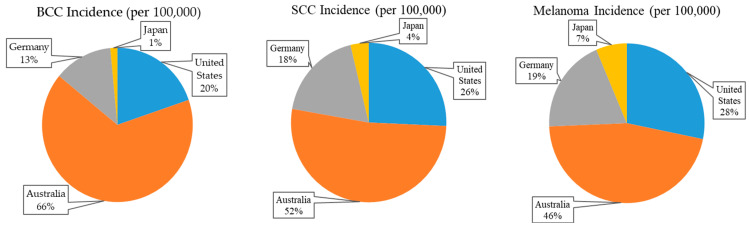
Global skin cancer incidence rates by country.

**Figure 3 gels-11-00033-f003:**
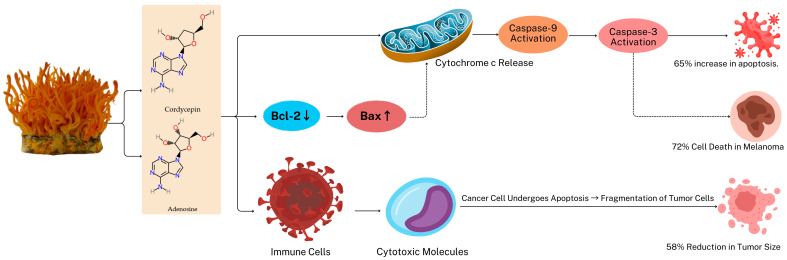
Mechanistic pathways of apoptosis induced by *C. militaris* compounds.

**Table 1 gels-11-00033-t001:** Bioactive compounds and anti-cancer mechanisms of natural ingredients.

Compound	Pharmacological Activities	Relevance in Research	Reference
Cordycepin	Anti-cancer, anti-inflammatory, immunomodulatory	Extensive research on anti-tumor and anti-inflammatory effects	[28]
Polysaccharides	Antioxidant, anti-inflammatory, immunomodulatory, antitumor	Frequently utilized in traditional medicine and modern formulations	[16,17,18,19]
Adenosine	Vasodilatory, anti-arrhythmic, anti-inflammatory	Emerging evidence for potential in dermatological applications	[35,36]
Sterols	Anti-inflammatory, anticancer	Potential therapeutic benefits in inflammation and cancer treatment	[37]
Cordymin	Antimicrobial, anti-inflammatory	Potential applications in combating microbial infections and reducing inflammation	[37]

**Table 2 gels-11-00033-t002:** Key bioactive compounds in *C. militaris* and their biological activities.

Compound	Concentration (mg/g)	Cancer Type	Research Model	Biological Activities	Reference
Cordycepin	91.78 ± 5.32	Melanoma	In vitro, in vivo	Induces apoptosis, cell cycle arrest	[53,54]
Polysaccharides	39.0–97.5	Squamous Cell Carcinoma	In vitro, in vivo	Immune modulation, anti-inflammatory, anti-metastatic	[55,56]
Adenosine	1.2 ± 0.14	Basal Cell Carcinoma	In vitro	Vasodilation, anti-inflammatory	[57,58]
Phenolic acids	32.27± 3.45	Melanoma	In vitro	Antioxidant, anti-aging	[59,60]

**Table 3 gels-11-00033-t003:** Integrated overview of *C. militaris* bioactive compounds and dermatological applications.

Compound	Mechanism of Action	Application	Observed Effects	Research Model	Reference
Cordycepin	Caspase activation, inhibition of DNA synthesis	Anti-cancer (melanoma, SCC)	65% increase in apoptosis, 72% cell death in melanoma	In vitro, in vivo	[54,73,74,75,76]
	NF-κB pathway inhibition	Anti-inflammatory, antioxidant	63% reduction in IL-6, 59% reduction in TNF-α	In vitro	
Polysaccharides	Stimulates immune cells, angiogenesis inhibition	Wound healing, barrier repair	43% increase in keratinocyte migration, 35% moisture retention	In vitro, in vivo	[55,56,68]
	Reduces oxidative stress	Anti-aging, UV protection	70% increase in SOD activity, 58% rise in CAT activity	In vitro	[71,77]
Adenosine	Modulates immune response, vasodilation	Anti-aging, sensitive skin	47% increase in collagen synthesis, 58% reduction in tumor size	In vitro	[78,79]
Phenolic acids	ROS scavenging, lipid peroxidation inhibition	Anti-aging, antioxidant	55% reduction in ROS levels, improved skin hydration	In vitro	[59,80]

**Table 4 gels-11-00033-t004:** Anti-inflammatory mechanisms of *C. militaris* and reduction in cytokine levels.

Compound	Mechanism	Targeted Inflammatory Marker	Reduction in Inflammatory Markers (%)	Reference
Cordycepin	NF-κB pathway inhibition	IL-6, TNF-α	63% reduction in IL-6, 59% in TNF-α	[117,123]
Polysaccharides	Immune modulation	IL-1β, IL-8	47% reduction in IL-1β, 52% in IL-8	[97,124]
Cordycepin	ROS suppression	Reactive Oxygen Species	66% decrease in ROS	[121,125]

**Table 5 gels-11-00033-t005:** Applications of *C. militaris* in skin cancer models.

Compound	Application Method	Dosage	Anti-Cancer Effect	Reduction in Tumor Growth (%)	Research Model	Reference
Cordycepin	Topical	1%	Inhibits cell cycle, promotes apoptosis	50%	Melanoma, in vivo	[133]
Polysaccharides	Systemic	20–40 mg/kg	Enhances immune response	45%	SCC, in vivo	[136]
Phenolic acids	Topical	2%	Reduces inflammation, UV protection	42%	Melanoma, in vitro	[27]

**Table 6 gels-11-00033-t006:** Anti-aging mechanisms of *C. militaris* and improvements in skin health.

Compound	Mechanism of Action	Dosage	Anti-Aging Effect	Improvement Observed (%)	Research Model	Reference
Cordycepin	ROS scavenging	0.50%	Reduces oxidative stress, preserves collagen	55%	Fibroblasts, in vitro	[140]
Polysaccharides	Hydration and GAG synthesis	2%	Enhances skin moisture retention	38%	Keratinocytes, in vitro	[142]
Adenosine	ERK pathway activation	1%	Stimulates collagen synthesis, improves elasticity	47%	Dermal cells, in vitro	[75]

**Table 7 gels-11-00033-t007:** Effects of *C. militaris* on wound healing and inflammation reduction.

Compound	Mechanism of Action	Dosage	Wound Healing/ Anti-Inflammatory Effect	Improvement Observed (%)	Research Model	Reference
Cordycepin	Enhances keratinocyte migration	0.20%	Accelerates wound closure, reduces scarring	43%	Wound model, in vitro	[151]
Polysaccharides	Moisturizing, barrier support	2%	Maintains hydration, promotes healing environment	35%	Skin explants, in vitro	[152]
Cordycepin	Inhibits TNF-α, IL-1β, IL-8	0.50%	Reduces redness and swelling	51%	Inflamed tissue, in vitro	[153]

**Table 8 gels-11-00033-t008:** Safety profile of *C. militaris* in dermatological applications.

Study Parameter	Findings	Incidence Rate (%)	Reference
General Skin Irritation	Minimal, mild irritation in some users	2%	[165]
Phototoxicity	No phototoxic effects	0%	[163]
Compatibility with Retinoids	No additional irritation observed	-	[130]
Allergic Reactions	Rare, mild cases of contact dermatitis	<2%	[160]

**Table 9 gels-11-00033-t009:** Comparative analysis of *C. militaris* and synthetic skincare agents.

Agent	Primary Function	Efficacy in Target Area	Typical Side Effects	Incidence of Side Effects (%)	Reference
*C. militaris*	Anti-aging, antioxidant	Reduces ROS by 40%, stimulates collagen	Mild irritation in sensitive skin	<5%	[178]
Retinoids	Anti-aging, collagen synthesis	High efficacy in cell turnover	Dryness, irritation, photosensitivity	~30%	[170]
Hydroquinone	Skin brightening	Reduces melanin effectively	Contact dermatitis, ochronosis	~10%	[173]
Ascorbic Acid	Antioxidant	Reduces ROS, brightening	Irritation in sensitive skin	~15%	[178]
BHT	Antioxidant	Preservative, moderate ROS reduction	Hormonal interference, irritation	~15%	[177]

**Table 10 gels-11-00033-t010:** Key areas for future research and standardization of *C. militaris*.

Area of Focus	Current Challenges	Recommended Actions	Reference
Clinical Trials	Limited human trials, variability in response	Conduct large-scale, controlled trials	[183]
Standardization of Extracts	Inconsistent bioactive compound levels	Develop industry-wide standard protocols	[186]
Quality Control	Label accuracy discrepancies (up to 30%)	Implement stricter regulations and testing	[187]

**Table 11 gels-11-00033-t011:** Suggested formulations of *C. militaris* for skincare applications.

Product Type	Primary Benefit	*C. militaris* Formulation	Complementary Ingredients	Reference
Anti-Aging Serum	Collagen support, elasticity	Cordycepin extract (1%)	Hyaluronic acid, vitamin C	[164]
Hydrating Cream	Moisture retention	Polysaccharide-rich *C. militaris*	Aloe vera, squalane	[65]
Brightening Serum	Skin tone improvement	Cordycepin + adenosine (0.5%)	Niacinamide, licorice root extract	[189]
Calming Gel	Redness reduction	Polysaccharide concentrate (2%)	Centella asiatica, calendula	[190]
Spot Treatment	Targeted anti-inflammatory	Cordycepin (0.2%)	Aloe vera, chamomile	[189]
Antioxidant Mask	Environmental defense	Whole *C. militaris* extract (3%)	Green tea, resveratrol	[65]
Overnight Cream	Skin regeneration	Cordycepin + polysaccharides (2%)	Peptides, ceramides	[43]
Eye Cream	Reduces fine lines	Adenosine (1%)	Caffeine, hyaluronic acid	[191]
Soothing Mist	Hydration, refreshes skin	*C. militaris* water extract (0.5%)	Rosewater, cucumber extract	[192]
Anti-Acne Lotion	Reduces inflammation, healing	Cordycepin (0.3%)	Tea tree oil, salicylic acid	[193]

## Data Availability

No new data were created or analyzed in this study.

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
