# Peer review of "Cordyceps militaris-Derived Bioactive Gels: Therapeutic and Anti-Aging Applications in Dermatology"

_gels, 2025, doi:10.3390/gels11010033_

Round 1
Reviewer 1 Report
Comments and Suggestions for Authors
This paper offers a comprehensive review of Cordyceps militaris and its bioactive compounds, which have shown promise in dermatology for their anti-aging, anti-inflammatory, and wound-healing properties. However, I have several suggestions for improvement:
- Figure 1: Please add the sources of the data, including the study years, and consider adding numerical values to clarify the tumor incidence percentages.
- Table 1: It would be helpful to specify whether the examples represent the most studied, most used, or most promising compounds in the literature.
- Introduction: Adding a concluding paragraph that outlines the main aim and structure of the review would give readers a clear roadmap.
- Bioactive Gels Section: The review title suggests a focus on Cordyceps militaris-derived gels, yet there is insufficient discussion on their formulation and applications. I recommend including a dedicated section on gels, distinguishing between individual active compound effects and their integration into gel matrices.
- Table 2: It would be useful to report concentrations as a range or provide an average concentration with standard deviation, along with more specific details about the cancer types and study models.
- Paragraphs 2 and 3: These paragraphs repeat the same information about cordycepin’s anti-inflammatory and anti-tumor mechanisms. Please consolidate the content to reduce redundancy and provide more specificity regarding the polysaccharides.
- Paragraph 3: This section lacks quantitative data, which would strengthen the argument. Consider revising the structure and focus of this paragraph, and perhaps dividing it into subsections.
- Tables: Several small tables can be merged for clarity and to provide more comprehensive information. For example, Table 3 and Table 4 could be combined into one.
- Table 5: This table could be transformed into a figure to visually illustrate the mechanisms, supported by more detailed biochemical pathways.
- Tables 7, 9, and 9 include percentages and effects without providing experimental conditions (e.g., dosage, duration, sample size). Please include this information for clarity.
Author Response
Dear Respective Reviewer,
We are deeply grateful for all your valuable comments to our manuscript. We agreed and revised our manuscript following your comments and suggestions. We send two files: a file is noted with track changes and color letters to indicate where we revised, and another file is the final revised manuscript without track changes. The responses to each your questions are detailed in the table below. Please kindly check.

Reviewer 2 Report
Comments and Suggestions for Authors
This paper reviewed the therapeutic potential of Cordyceps militaris derived bioactive gels in dermatology and skincare, emphasizing their anti-aging and skin cancer benefits. At the same time, the review underscores the importance of clinical trials and standardization to ensure safety, efficacy, and consistency in applications, paving the way for C. militaris-based gels as an innovative approach in skincare and dermatology.
Overall, this review paper is well organized. The sections' design and description are scientific enough. Minor revision is needed. I would accept this review paper if the authors can solve the following issues.
1. In the first two paragraphs of the introduction section, the authors displayed too much information about skin cancer, including the skin cancer classification. However, the authors did not compose the main text based on the therapeutic of Cordyceps militarist on cancer classification. I would suggest simplifying the description of skin cancer.
2. The introduction section needs a concluding paragraph to summarize the main content of this review.
3. I suggest adding a paragraph on how to extract (the method to extract) the Bioactive Compounds from Cordyceps militaris in section 2.
4. It is highly recommended to cite this reference (10.1002/adma.202400310) in lines 130-132, supporting that IL-6 and TNFa can be used as an indicator of skin inflammation.
Author Response

(The authors gave the same response as above.)

Reviewer 3 Report
Comments and Suggestions for Authors
This manuscript "Potential Applications of Cordyceps militaris-Derived Bioactive Gels in Dermatology and Skincare: A Focus on Therapeutic and Anti-Aging Properties" explored the potential of C. militaris-derived bioactive gels for applications in dermatology and skincare, with a particular focus on their therapeutic and anti-aging benefits. Overall, this article is good and covers important area. These are the suggested corrections:
1- The title is too long and better to be shortened.
2- I suggest the authors to make a graphical abstract.
3- There is no even single figures or cartoon. The addition of relevant figures and authors cartoon explaning the findings and where the filed is moving forward will increase the quality of article.
4- As this is review article, the authors prospective and opinion needs to be included in this review.
5- I think it is better to make an abbreviation list and include it in the manuscript.
6- The subsections needs to further classified into the sub topics like, based on type of dosage form.
7- Proof-reading is required for this manuscript.
Comments on the Quality of English LanguageMinor editing.
Author Response

(The authors gave the same response as above.)

Round 2
Reviewer 1 Report
Comments and Suggestions for Authors
the paper is definitely improved. Thanks.
Reviewer 3 Report
Comments and Suggestions for Authors
The authors have addressed the comments successfully.